# Diagnosis and Treatment of Myogenous Temporomandibular Disorders: A Clinical Update

**DOI:** 10.3390/diagnostics12122914

**Published:** 2022-11-23

**Authors:** Natalie Hoi Ying Chan, Ching Kiu Ip, Dion Tik Shun Li, Yiu Yan Leung

**Affiliations:** 1Faculty of Dentistry, The University of Hong Kong, Hong Kong; 2Oral and Maxillofacial Surgery, Faculty of Dentistry, The University of Hong Kong, Hong Kong

**Keywords:** temporomandibular disorders, temporomandibular joint dysfunction syndrome, facial pain, temporomandibular joint, myalgia

## Abstract

Myogenous temporomandibular disorders (M-TMDs) are the most common chronic orofacial pain, affecting the masticatory muscles and, thus, jaw movement. While a concise diagnosis is crucial to formulate a rational treatment plan, the similarities in clinical presentations that M-TMDs share with other neuromuscular disorders affecting the temporomandibular joint (TMJ) could easily confuse physicians. In addition to the basics, such as thorough history taking and meticulous clinical examinations, different imaging techniques are useful adjuncts to facilitate the diagnostic process. This review presents an overview of the current understanding on a variety of diagnostic and treatment modalities for M-TMD patients. It is essential to highlight that there is not a single treatment for all, and the benefits of multidisciplinary strategies have been noted for the effective management of myogenous TMD pain. Treatment modalities ranging from conservative to minimally invasive options are discussed in this review.

## 1. Introduction

Temporomandibular disorders (TMDs) refer to a heterogeneous group of musculoskeletal dysfunctions affecting the temporomandibular joint (TMJ) and/or the masticatory muscles [1] that control jaw movement. According to Diagnostic Criteria for TMD (DC/TMD) Axis I, TMDs are divided into Group I: muscle disorders (including myofascial pain with and without mouth-opening limitation; Group II: involving disc displacement with or without reduction and mouth-opening limitation; and Group III: arthralgia, arthritis, and arthrosis [2]. They are the most common chronic orofacial pain, affecting 31% of adults and 11% of children and adolescents among the general population [3,4] TMDs and myogenous temporomandibular disorders (M-TMDs), also known as masticatory myalgia, are the most common condition, affecting 45.3% of TMD cases [2,5,6,7,8].

While there has been extensive research on TMD, the pathophysiology is not completely understood. TMDs have a multifactorial aetiology, and among them, some researchers reported that central sensitisation may play a role in chronic pain in M-TMD patients. Contrary to arthrogenous TMD, which appears to be a localized phenomenon, myogenous TMD may present overlapping features with other disorders, such as fibromyalgia and primary headaches, characterized by chronic primary pain related to dysfunction of the central nervous system (CNS), probably through the phenomenon of central sensitisation. Thus, central sensitisation could represent the basis of chronic pain, “or pain that persists beyond a normal time of healing” in patients affected by TMD [9]. However, TMDs of myogenous origin are generally considered to be strongly associated with psychogenic factors such as psychological stress, anxiety, depression, sleep and hormonal disturbances [10]. Some researchers reported that patients diagnosed with myofascial pain have more severe depressive and nonspecific physical symptoms than patients diagnosed with TMJ internal derangement (i.e., disk displacement) [11,12]. Factors like facial asymmetry and other forms of dentofacial deformities are proven to be correlated with TMD because of imbalanced masticatory forces, while the correction of these deformities could bring improvements to the TMD symptoms [13,14,15]. Recent studies have also suggested that during periods of ongoing lockdown and isolation due to the COVID-19 pandemic, people who were frequently under stress and anxiety may be more likely to develop depression and TMD symptoms [16]. One theory holds that these people may be more likely to clench their muscles [8], a form of repeated strain that leaves muscles susceptible to myofascial trigger points (MTrPs) [17,18]. Moreover, patients with more severe signs and symptoms of TMD had a lower pressure pain threshold [19]. These findings that psychological variables are closely tied to the development of TMD have been confirmed by the Orofacial Pain: Prospective Evaluation and Risk Assessment (OPPERA) study [20].

Intriguingly, sleep bruxism was also found to have a positive correlation with myofascial pain, arthralgia and joint pathology, such as disc displacement and joint noises. In articles that used polysomnographic analysis (PSG) for bruxism diagnosis, a positive correlation was established between bruxism and masticatory muscular activity of the temporal and masseter muscles during sleep. It is demonstrated that the increase in EMG activity during sleep could be a risk factor for myofascial pain. Thus, it is possible to suggest that bruxism would be associated with TMD [21].

Despite the research effort on M-TMD up until now, establishing a correct diagnosis which is critical for the selection of the proper treatment remains a challenge for physicians. This is because the presentation of M-TMD may mimic other non-TMD conditions and requires a more comprehensive patient assessment. At present, there is no simple one-step diagnostic test to help pinpoint a definitive diagnosis of M-TMD.

This article aims to provide an overview of the current diagnostic and treatment modalities available in the management of M-TMD. Both conservative and minimally invasive options will be discussed, as there is not a single treatment for M-TMD which encompasses a wide range of diagnoses.

## 2. Diagnoses and Classifications

According to the new evidence-based Diagnostic Criteria for TMD (Axis I DC/TMD protocol) [2], muscle pain myalgia represents what was called myofascial pain in RDC/TMD. The term myofascial pain now describes two new DC/TMD diagnoses: myofascial pain and myofascial pain with referral.

For the new DC/TMD classification, myogenous TMD pain diagnoses are organized into four major subclasses: myalgia, tendonitis, myositis and spasm (Table 1). Myalgia is further subdivided into local myalgia, myofascial pain and myofascial pain with referral upon myofascial examination protocol (Table 2).

Myalgia is described as pain of muscle origin that is affected by jaw movement, function or parafunction, and replication of this pain occurs with provocation testing of the masticatory muscles. Patients with TMD will experience pain in the jaw, temple, ear or front of ear and pain modified with jaw movement, function or parafunction. This is acute to a chronic condition that includes the presence of regional pain associated with tender areas referred to as trigger points (TrPs), which are expressed in taut bands of skeletal muscles, tendons or ligaments [22]. Although the pain occurs most often in the region over the TrPs, pain can be referred to areas distant from the TrPs (e.g., temporalis, referring to the frontal area, and masseter, referring to the ear or the posterior teeth). Often, reproducible duplication of pain complaints with specific palpation of the tender area is diagnostic.

The diagnostic algorithms in the new DC/TMD for myalgia now include criteria for modification of pain by function, movement or parafunction; these criteria are also included in the TMD Pain Screener [23]. Currently, the clinical examination for myalgia includes pain with jaw-opening movements and palpation of the temporalis and masseter muscles. Pain from these provocation tests must replicate the patient's pain complaint. To differentiate the three types of myalgia, the duration of the 1 kg of palpation pressure is increased to 5 s to allow more time to elicit spreading or referred pain, if present. Pain is localized during palpation (local myalgia) or spreads within (myofascial pain) or beyond (myofascial pain with referral) the palpated muscular territory. If a diagnosis of myalgia is desired and no distinction between the three types is needed, the disorder of myofascial pain with a limited opening, as described in the RDC/ TMD, is eliminated.

Although tendonitis, myositis and spasm were less routinely encountered, it is important to include them in the differential diagnosis of TMJ disorders and pains. In addition, problems such as neoplasms, migraine, neuralgia and psychological disorders should also be considered. One case-control study [24] found that the diagnosis of myofascial pain is significantly higher in fibromyalgia patients. According to this line of thinking, one must not forget that TMD symptoms, which are difficult to diagnose and often missed, should be kept in mind in the management of fibromyalgia patients.

## 3. Diagnostic Approaches

Given the multifactorial aetiological nature of TMDs, a thorough history and clinical examination remain the cornerstones for the diagnosis of TMD [8].

Assessment of TMD patients should begin with a history taking of pain which follows the same format for other pain conditions [25]. Common chief complaints include pain on chewing/mouth opening, headache, ear pain, lack of chewing force, easily induced fatiguability in the masticatory muscles and disability to open the mouth wide [26]. Patients with myogenous TMD usually describe the condition to be a spontaneous dull aching pain and localized tenderness or stiffness in the masticatory muscles. A visual analogue scale (VAS) should be used to grade the severity of pain, so treatment progress can be quantitatively monitored. A past and current medical history, including a full medications list, may reveal any comorbidities that may be related to TMD. The clinician should also pay attention to any habits such as smoking, drinking and recreational drug use, and any history of clenching or bruxism as reported in complaints by the patient’s bed partner. Additionally, the clinician should ask questions regarding stress levels or the presence of psychiatric illnesses such as anxiety and depression, as they are consistently associated with TMD. Although most clinicians treating TMD are experienced in obtaining a clinical history, some may not be comfortable with taking a psychological history. If desired, the clinician may employ numerous psychosocial instruments available to aid in their diagnosis, such as those in Axis-II of DC/TMD [2]. When necessary, the patient may be referred for a psychological evaluation.

Diagnosis of myogenous TMD relies largely on physical palpation of the soft tissue by a trained physician [27] and the recognition of pain points by the patient. The confirmation of the location of pain in the masseter and temporalis muscle, and whether the pain is confined, remains within or spreads beyond the confines of the muscle, should be confirmed with a muscle and MTrP palpation, which is carried out at rest and during mandibular function. A steady firm pressure (~2 kg) should be applied firmly over the muscle of concern when in the relaxed state for at least 1–2 s, and the patient should be asked to rate the tenderness during the palpation. Palpation should follow the direction of the muscle fibres to detect taut bands and check for the presence of MTrPs which produce referred pain to a nearby site.

Maximum mouth opening (MMO), including pain-free maximum mouth opening, maximum unassisted mouth opening and maximum assisted mouth opening, should be assessed in each appointment [28]. Any pain with maximum unassisted or assisted opening should also be noted.

### 3.1. Imaging Modalities

Ideally, the diagnosis of TMD pain is reached by a combination of clinical manifestations and diagnostic imaging confirmation. When further imaging is desired, CBCT and MRI are the diagnostic imaging techniques most commonly used in the field of dentistry to aid the diagnosis of TMD [29]. While CBCT is optimal for viewing skeletal and dental tissues which are especially useful in identifying degenerative joint diseases such as osteoarthritis [30], MRI is considered to be the gold standard when assessing the articular disc in terms of location and morphology [29,31] as MRI can provide early detection of disc abnormalities and the presence of joint effusion [32]. All disc-related TMD problems can be confirmed by MRI when indicated [31].

Regarding TMD with myogenous origin, whereas MTrPs are one of the most common and important causes of musculoskeletal pain, detection of MTrPs is critical for more comprehensive clinical evaluation and treatment of TMD patients. Unfortunately, CBCT and MRI are not able to detect the presence of taut bands or MTrPs. Nevertheless, when further information is desired, for example, the correct localization of MTrPs needed for different pain relief techniques, notably dry needling and injection-based therapies, ultrasonography may be a viable tool for locating the taut band or MTrPs [33].

Diagnostic ultrasound (US) has been proposed as a method to improve the reliability of MTrPs’ localization as it is relatively cheap and accessible. Although the diagnostic efficacy of US is highly dependent on the operator’s skills, knowledge of compartmental muscle anatomy and experience in assessing normal and abnormal muscle tissue in the static and dynamic state [34], many articles have shown that US could identify MTrP on upper trapezius muscle or other musculature [33,35,36,37]. Yet, little evidence was found for muscles of mastication. Our centre is now conducting a clinical trial on patients presented with myogenous TMD to assess the diagnostic efficacy of US, and its findings will be presented in due course.

### 3.2. Diagnosis of M-TMD

The International Association for the Study of Pain Subcommittee on Taxonomy [38] has classified myofascial pain as pain in any muscle with MTrPs that are very painful upon compression during palpation and cause referred pain.

Myofascial pain is diagnosed in five scenarios according to Shah et al. [27]: (1) when the clinician feels a hyperirritable spot within a palpably taut band of muscle fascia; (2) upon sustained compression of this hyperirritable spot, the patient reports new or increased dull aching pain in a nearby site; (3) when a decreased range of unassisted movement of the involved body area is noted; (4) weakness without atrophy and no neurological deficit explaining this weakness; (5) the presence of referred autonomic phenomena upon compression of the hyperirritable spot and/or a twitch response to snapping palpation of the taut bands.

## 4. Treatment Modalities

The treatment approach of TMD can be broadly classified into three types: conservative, minimally invasive and invasive [39]. Currently, the paradigm has shifted from open procedures to non-invasive options [8,40]. Thus, reversible conservatory approaches are usually considered the first line of treatment [8,41]. As surgical approaches usually target arthrogenous TMD, they are not the focus of this article. Several treatment modalities have been reported to successfully treat M-TMD by pain relief and restoring mouth opening. Nonetheless, to date, there is still a lack of consensus and evidence as to which is the most preferred option [42,43,44,45].

### 4.1. Conservative Therapies

#### 4.1.1. Pharmacological Therapy

Medications are often prescribed initially as a non-invasive measure to treat TMD [8,46]. The most commonly used pharmacological agents are non-steroidal anti-inflammatory drugs (NSAIDs), muscles relaxants and anti-depressants. Despite carrying some well-known adverse effects, especially gastrointestinal disturbance, the anti-inflammatory and analgesics properties of NSAIDs, for example, ibuprofen, naproxen, diclofenac etc., render them popular among clinicians in the management of TMD, and these are sometimes considered as the first-line drugs of choice [46,47,48]. Besides, NSAID topical ointments are available as an alternative to lessen systemic absorption [40]. In a systematic review with meta-analysis conducted in 2017, Haggman et al. acknowledged the positive treatment effect of the muscle relaxant cyclobenzaprine for M-TMD, despite a lack of understanding of their long-term use and associated side effects [49]. Structurally similar to cyclobenzaprine are tricyclic antidepressants (TCAs). At a relatively low dose compared to treating depression, TCAs have been shown to be effective in reducing the frequency and intensity of pain arising from TMD; however, the numerous detrimental effects that ensue, notably dry mouth, fluid retention and cardiotoxicity, have limited its routine use [50]. Another potent class of central-acting drugs are opioids; however, their use has been discouraged to minimize central nervous system depression and physical dependence [51].

Although various medications are used in the management of TMD, there has been, generally, a paucity of evidence to support a standardized regimen and the best class of medications to be used [40,52]. Therefore, the clinical decision still lies mainly with the clinicians’ experience and comfort [53], tailored to the patient’s best individual needs.

#### 4.1.2. Occlusal Splint

The use of occlusal splints has been reported to have improved the mandibular movement and pain for patients with TMD [39,43,54,55,56] and is considered a basic treatment for TMD [44,54,57,58]. Yet, there has been, generally, a scarcity of strong evidence to validate its efficacy in these aspects [42,59], especially in the long term [57,60,61]. Interestingly, Alkhytari et al. have conducted a systematic review on stabilization splints in 2018 and concluded that the patient-reported treatment satisfaction, including domains other than pain relief such as psychological well-being, was beyond that of a placebo effect. The authors also suggested that, apart from the pain scale, variables concerning patient-reported satisfaction should also be considered when evaluating the efficacy of different treatment modalities [62].

Different splint designs are available at present; the most commonly used ones are soft or hard stabilization splints, including Tanner appliance, Fox appliance, Michigan splint or centric relation appliance, anterior repositioning splint and anterior bite splint. When compared with TMD of joint origin, hard stabilization splints have been found to yield better outcomes in treating myogenous TMD [63]. They can aid in promoting the functional recovery of masticatory muscles [64] and restoring postural balance [65]. While mini-anterior splints (similar to anterior deprogramming splints in mechanism) have been suggested to be the most effective splint design to manage muscle pain in TMD, there is still a lack of high-level evidence [63]. On the other hand, its long-term use is not always advocated due to the possible adverse effects on occlusal stability from prolonged disocclusion of posterior teeth [66]. Moreover, some splints that incorporate biofeedback features, such as vibratory stimulus upon parafunctional occlusal load, have been reported to offer additional treatment benefits [67].

#### 4.1.3. Physiotherapy

Physiotherapy, also termed physical therapy or exercise therapy, has been proposed to play a vital role, especially in the management of myofascial pain complaints for patients with TMD [8,48,68,69,70,71,72]. It can be subdivided into self-exercise conducted by patients at home or manual therapy by a trained practitioner [73]. Educations on various homecare strategies, for instance, massage of the masticatory muscles, jaw opening exercise and applications of moist heat pads [69,73] are most commonly offered as low-cost, useful modalities free of adverse consequences at an early phase. Evidence has also attached importance to postural correction of the head and neck in reducing pain and increasing jaw mobility [70,74], which might be related to the restoration of cervical lordosis [75]. Furthermore, some widely cited examples of manual therapies are post-isometric muscle relaxation and myofascial release. Although the aforementioned procedures have been more extensively documented in treating muscles of the trunk and limbs, these relaxation techniques have been reported to improve musculoskeletal functions even in the masticatory system, possibly by relief of muscle tension [76]. A diversity of approaches are available and have been shown to be beneficial in the treatment of TMD, including but not limited to mobilization, stretch, endurance exercise, etc., but as of today, none has been proven superior; more scientific evidence is called for to formulate a standardized protocol [73,77,78,79]. Additionally, there has been evidence suggesting that physiotherapy might be able to improve headache-associated symptoms from TMD [69].

#### 4.1.4. Electrical Modalities

Apart from self-manipulation therapy, electrical modalities are becoming a more popular non-invasive treatment modality for relieving acute and chronic pain in TMD patients.

A. Transcutaneous Electrical Nerve Stimulation (TENS)

Transcutaneous electric nerve stimulation (TENS), has been used for millennia to relieve pain. It utilizes electrodes placed on the skin, which are connected to the unit via wires to achieve a targeted therapeutic goal [80] (Figure 1). Electrical impulses are generated to descend pain signals to the spinal cord and brain, stimulate the production of endorphins, relieve peripheral and neuropathic pain and relax muscles [81]. Not only pain and muscle tenderness can be relieved [82]; TENS also demonstrates benefits in improving masticatory function in TMD patients by improving their mouth opening and eventually increasing their biting force [83]. As seen from the photo, TENS is a small and portable device, often battery-operated, which can sometimes even fit into a pocket. Another additional benefit of TENS is that patients can self-apply the electrical pads themselves without assistance.

B. Low-Level Laser Therapy (LLLT)

Among the various physical therapy modalities, low-level laser therapy (LLLT) has been placed under the spotlight because of its non-invasive, safe, easy application and short treatment time [84]. A recent systematic review has suggested that laser therapy has been particularly useful to treat muscle-related TMD pain among other rehabilitative approaches [85]. In addition, also termed photobiomodulation (PBM), LLLT involves a light source that emits no heat, sound or vibration but could affect the function of fibroblasts, facilitate repair and act as an anti-inflammatory agent [86]. One special feature of LLLT is that it does not make contact with skin and can be used even with wounds. In a systematic review and network meta-analysis conducted in 2022, Ren et al. explored the optimal wavelength range of laser application, affirming that laser therapy with a wavelength of 910–1100 nm was the most effective [87].

C. Therapeutic Ultrasound (US)

Similar to other electrical modalities, therapeutic US introduces energy to tissue cells, aiming to improve circulation to tissues and facilitate the healing process (Figure 2). There are two modes in therapeutic modes: continuous mode, which produces a deep heating effect (for chronic pain); and intermittent mode, which will not increase tissue temperature (for acute pain). US therapy can significantly reduce the pain and improve the functionality of the temporomandibular joint and mouth-opening limit for TMD patients in four weeks [88]. Yet, 2.63% of patients that had undergone ultrasound therapy had relapse and recurrence of pain [88]. Therefore, its long-term effectiveness is still inconclusive.

While TENS, LLLT and therapeutic US are useful in pain reduction, systematic reviews and meta-analyses carried out in 2022 have shown that LLLT was the most effective in reducing pain among the three treatment modalities [87,89,90]; LLLT was found to be superior to TENS and was also proven to be better in reducing pain than therapeutic US [89]. Better results could be achieved with higher wavelengths, and wavelengths ranging from 910 nm to 1100 nm were recommended to treat TMD using LLLT [87].

D. Extracorporeal Shockwave Therapy (ESWT)

Radial and focused extracorporeal shockwave therapy (ESWT) is gaining popularity for treating musculoskeletal cases. It was hypothesized that the main biological effect on tissue treated by ESWT is an increase in the permeability of cell membranes and the release of several molecules stimulating tissue regeneration [91], such as vascular endothelial growth factor (VEGF), fibroblast growth factor (FGF) and the activation of the endothelial nitric oxide synthase (eNOS) with angiogenic effects [92]. Most importantly, it is believed that ESWT can modulate the release of anti-inflammatory mediators and endorphins that activate descending inhibitory system to relieve pain [93].

In 2022, Marotta et al. conducted an RCT to evaluate the efficacy of physical exercise, with or without radial extracorporeal shock wave therapy (rESWT), in patients with only muscular TMD. The findings of this pilot RCT suggested that rESWT combined with physical therapy could be effective in relieving pain and improving function in muscle-related TMD patients [94]. There was another interesting discovery from a qualitative analysis in 2022, which concluded that ESWT could facilitate both clinical and functional recovery in people with myofascial pain syndrome, but not for fibromyalgia [95]. Another point to note is that extracorporeal radial shockwave therapy combined with ultrasound-guided injection of lidocaine into MTrPs has been shown to be more effective for reducing pain and elastic stiffness in myofascial pain syndrome in the fourth week [96].

Despite considerable research that has supported the efficacy of ESWT on MTrPs in the trapezius muscle [97,98,99,100], little is known regarding its effect on mastication muscles. Considering this research gap, more clinical studies in this area are encouraged. Currently, our centre is conducting a randomized clinical trial on its use in patients presenting with myogenous pain (Figure 3); clinical improvements have been confirmed in our pilot study and its findings will be presented when the study is completed.

#### 4.1.5. Psychological Intervention

Emerging evidence has shed light on the psychological component in both the aetiology and management of TMD [101]. There was an interesting finding by Nifosi et al. that patients suffering from M-TMD usually reported a higher stress level than those articular patients [102]. While it is logical to assume the causative relationship between anxiety and parafunctional behaviours, which could contribute to muscle hypertrophy [45], there are still insufficient data in the literature to elucidate the exact pain–psychopathology link. Nonetheless, in light of the multifactorial nature of TMD [45], psychosocial assessment has been integrated into its treatment to promote patients’ mental well-being and potentially reduce harmful habits [103], leading to a rise in the popularity of a multimodal, biopsychosocial approach [63,103,104,105]. To deliver a comprehensive psychological-based therapy, a joint effort between TMD specialists and psychologists might be required; one notable, evidence-based example is cognitive behavioural therapy (CBT) [106,107,108].

However, more often than not, dental professionals might not be equipped with a solid background in psychological domains; therefore, counselling can be offered in the form of patient education, for example, on normal jaw function, suspected aetiological factors and reassurance of its benign nature, strategies with proven clinical efficacy [109,110,111,112].

### 4.2. Minimally Invasive Treatment

#### 4.2.1. Dry Needling/Acupuncture

Dry needling (DN) or acupuncture are both treatment strategies targeted at the muscles, which have been widely used to treat a myriad of neuromusculoskeletal diseases, including myogenous TMD [113]. Although they differ slightly in their philosophy, western-based DN and acupuncture originating from traditional Chinese medicine both involve the insertion of long and fine needles into the MTrPs to relieve muscle tension and produce an analgesic effect [114,115,116]. Another distinguishing difference is that during DN, TrPs are repeatedly perforated internally and externally with the needle, such that a local twitch response might sometimes be observed [117]. Whilst insufficient data are available to compare the efficacy of these two needling techniques [118], they have been useful aids to manage pain and restore motions in some patients with muscle-originated TMD [48,114,119,120], despite the lack of clarity regarding the mechanism they are based upon [114]. However, due to the low quality of evidence and heterogenicity of the studies conducted, needling therapies have not been regarded as a first-line treatment for M-TMD [116,118].

#### 4.2.2. Minimally Invasive Injections

A variety of drugs are available for injection therapy in the management of TMD [46]. They are usually classified by the mode of delivery: either as intra-articular injection, into the TMJ alone or as part of an arthrocentesis procedure [121,122]; or into the mastication muscles [123], and this article shall focus on the latter. Usually targeted at the MTrPs, intramuscular injections are also termed trigger point injections (TPIs). They can involve the use of local anaesthetic substances such as lidocaine, corticosteroids or botulinum toxin (BTX) [124]. In the past, lidocaine seems to be the preferred option for TPI due to its low cost [125]. Although limited information is available in the literature to compare the efficacies of different injectates, it has been observed that BTXs are increasingly popular in recent years [126], despite a lack of consensus on its clinical value [127,128,129,130]. For selected patients, especially those refractory to conventional conservative treatment measures, BTX injection might be a useful tool in modulating pain threshold and restoring motion [46,131,132,133,134,135,136,137,138]. It has been noted that the administration of BTX can also improve sleep bruxism [139], which has long been closely associated with the signs and symptoms of myogenous TMD. Note that the therapeutic effect of BTX injection is usually transient and repeated appointments are expected for long-term relief [140,141,142]. Concerns over side effects such as muscle paralysis and financial implications [143,144] have rendered it no more than an adjunct to other standard treatments [8].

## 5. Conclusions

Branched off from the umbrella term TMD, masticatory myalgia shares equally bewildering aetiology with multifaceted signs and symptoms. Aside from somatic cause, current evidence has recognized the role of psychosocial factors in its course of development. With this in mind, contemporary treatment approaches have placed more emphasis on bio-behavioural interventions, such as counselling therapy, alongside simultaneous conservative measures, to address various aspects of the issue in a multimodal fashion.

Note that this article is limited by the absence of meticulous meta-analysis in a systematic manner. It is our objective to provide an updated narrative overview of diagnosis and treatment modalities available for M-TMD. It has been widely accepted that the treatment philosophy of TMD remains empirical due to a paucity of knowledge in its pathophysiology. Further studies are needed to make sense of the clinical conundrum.

## Figures and Tables

**Figure 1 diagnostics-12-02914-f001:**
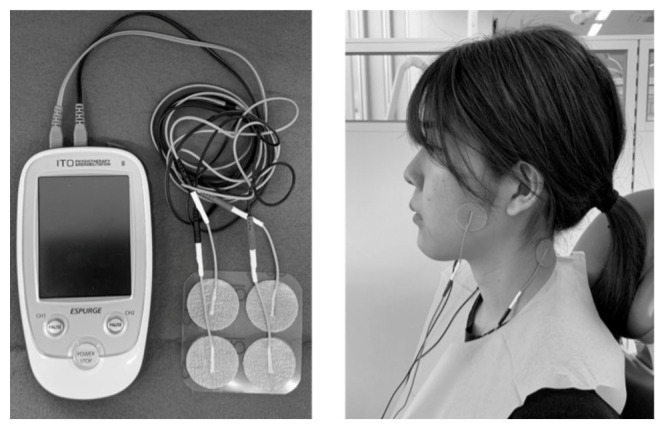
The transcutaneous electric nerve stimulation (TENS) device and its application on the masticatory muscles. (Figure from [83], https://pubmed.ncbi.nlm.nih.gov/33081336/#&gid=article-figures&pid=figure-1-uid-0 (accessed on 18 November 2022)).

**Figure 2 diagnostics-12-02914-f002:**
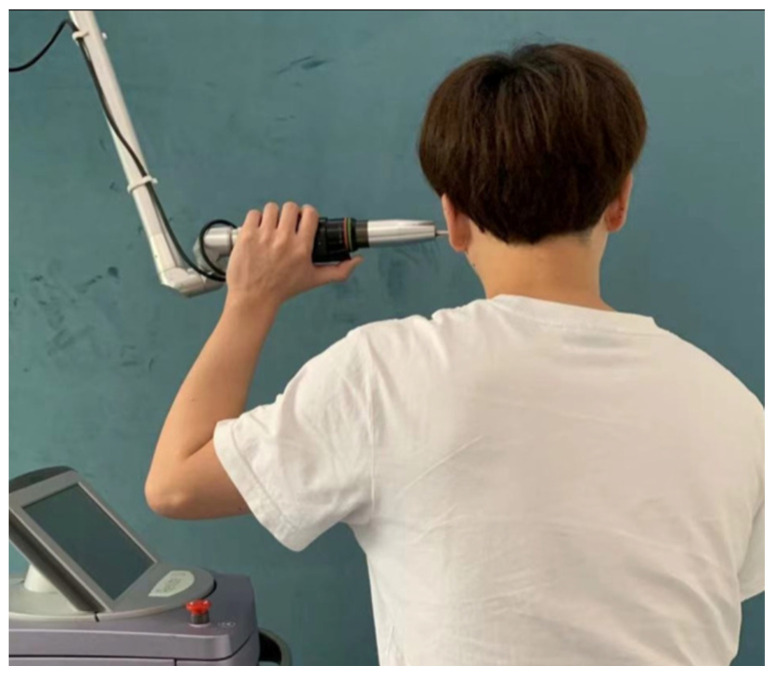
Therapeutic ultrasound (US). (Figure from [88], https://pubmed.ncbi.nlm.nih.gov/34140803/#&gid=article-figures&pid=figure-1-uid-0 (accessed on 18 November 2022)).

**Figure 3 diagnostics-12-02914-f003:**
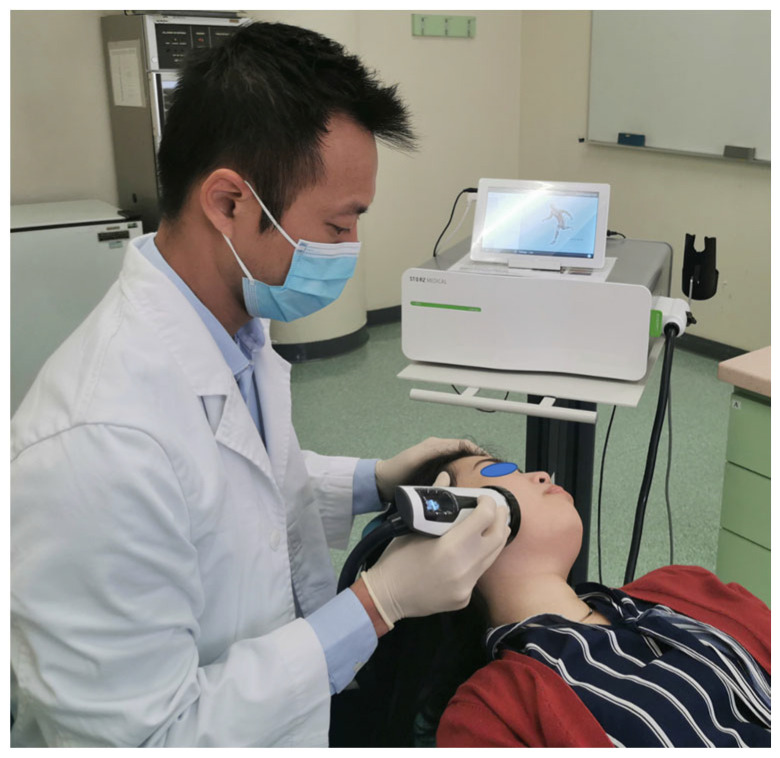
ESWT shows promising results for the treatment of MTrPs of masseter muscles in patients with myogenous TMD.

**Table 1 diagnostics-12-02914-t001:** Differential diagnosis of M-TMDs according to the DC/TMD classification.

Differential Diagnosis of M-TMD
Muscle pain Myalgia Tendonitis Myositis Spasm
2. Contracture
3. Hypertrophy
4. Neoplasm
5. Movement Disorders Orofacial dyskinesia Oromandibular dystonia
6. Masticatory muscle pain related to central/systemic pain disorder Fibromyalgia/widespread pain

**Table 2 diagnostics-12-02914-t002:** Subdivision of myalgia according to the DC/TMD classification.

Classification of M-TMD	Clinical Findings
Myalgia	Familiar pain in the masseter and temporalis upon palpation or mouth opening
Local myalgia	Familiar pain in the masseter and temporalis localized to the site of palpation
Myofascial pain	Pain in the masseter and temporalis spreading beyond the site of palpation but within the confines of the muscle being palpated
Myofascial pain with referral	Pain in the masseter and temporalis beyond the confines of the muscle being palpated

## Data Availability

Not applicable.

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
