# Peer review of "Diagnosis and Treatment of Myogenous Temporomandibular Disorders: A Clinical Update"

_diagnostics, 2022, doi:10.3390/diagnostics12122914_

Round 1
Reviewer 1 Report
I suggest the authora to write in the chapter related to the treatment about the occlusal splints to speak also about the 3d splints.
The authors could take informations from this article
- DOI:
- 10.37358/RC.18.11.6688
Author Response
We sincerely thank the reviewer for their time and effort to review our paper.
Indeed, 3D printing is an emerging trend worldwide and in dentistry that provides an important and accurate alternative to the traditional way of fabrication. However, we think that the type of splint design is more pertinent to clinicians in their decision-making process in managing patients with myogenous TMD, therefore, we have decided to keep our focus and leave the topic for future discussion. Once again we thank you for your suggestion and the provision of references.
Reviewer 2 Report
Dear Authors,
The aim of this article is to provide an overview of the current diagnostic and treatment modalities available in the management of muscular-related temporomandibular disorders (M-TMD). Both conservative and minimally invasive options will be discussed as there is not a single treatment for M-TMD, which encompasses the wide range of diagnoses.
The study was in line with the aims of the journal.
However, there are some issues that should be addressed.
The study was in line with the aims of the journal.
However, there are some issues that should be addressed.
Abstract
“While a rational treatment 11 design is based on a concise diagnosis, the similarities in clinical presentations that M-TMD share 12 with its other neuromuscular counterparts affecting the temporomandibular joint (TMJ) constitute 13 a challenge for physicians to make sound clinical decisions”. This sentence is not clear.
Introduction
I suggest improving the introduction section.
After definition, please report the classification according to Diagnostic Criteria for TMD (DC/TMD) Axis I. Thus, report that TMD could be divided in Group I:muscle disorders (including myofascial pain with and without mouth opening limitation); Group II: including disc displacement with or without reduction and mouth opening limitation; Group III: arthralgia, arthritis, and arthrosis.). (cite and refer to: Schiffman E, Ohrbach R, Truelove E, et al. Diagnostic Criteria for Temporomandibular Disorders (DC/TMD) for Clinical and Research Applications: recommendations of the International RDC/TMD Consortium Network* and Orofacial Pain Special Interest Group. J Oral Facial Pain Headache. 2014;28(1):6-27.).
Among causes, report that central as factors involved in TMD. Compared to arthrogenous TMD, which appear to be a localized phenomenon, myogenous TMD may present overlapping features with other disorders, such as fibromyalgia and primary headaches, characterized by chronic primary pain related to dysfunction of the central nervous system (CNS), probably through the phenomenon of central sensitization. Thus, the central sensitization could represent the basis of chronic pain, “or pain that persists beyond a normal time of healing” in patients affected by TMD (please cite and refer to: Pain Management and Rehabilitation for Central Sensitization in Temporomandibular Disorders: A Comprehensive Review. Int. J. Mol. Sci. 2022, 23, 12164. doi: 0.3390/ijms232012164).
I suggest also to report sleep bruxism, its diagnosis and its rule in TMD (Jiménez-Silva A, Peña-Durán C, Tobar-Reyes J, Frugone-Zambra R. Sleep and awake bruxism in adults and its relationship with temporomandibular disorders: A systematic review from 2003 to 2014. Acta Odontol Scand. 2017 Jan;75(1):36-58. doi: 10.1080/00016357.2016.1247465.)
4. Treatment Modalities
Refer to a recent systematic review with meta-analysis that concluded that conservative approaches were effective in reducing muscle-related pain in TMD patients. Indeed this research included only RCTs evaluating patients with myogenous TMD (only Group I according to DC/TMD). Cite and refer: Efficacy of rehabilitation on reducing pain in muscle-related temporomandibular disorders: A systematic review and meta-analysis of randomized controlled trials. J Back Musculoskelet Rehabil. 2022;35(5):921-936. doi: 10.3233/BMR-210236. PMID: 35213347.
B. Low Level Laser Therapy (LLLT)
In 2022, Ren et al. conducted a systematic review and network meta-analysis aimed at exploring the optimal wavelength range of laser application, affirming that laser therapy with wavelength of 910-1100nm was the most effective (cite and refer to: Comparative effectiveness of low-level laser therapy with different wavelengths and transcutaneous electric nerve stimulation in the treatment of pain caused by temporomandibular disorders: A systematic review and network meta-analysis. J Oral Rehabil. 2022 Feb;49(2):138-149. doi: 10.1111/joor.13230. Epub 2021 Aug 21. PMID: 34289157.)
D. Extracorporeal Shockwave Therapy (ESWT)
In 2022 Marotta et al condicted a RCT to evaluate efficacy of physical exercise, with or without radial Extracorporeal Shock Wave Therapy (rESWT), in patients with only muscular TMD. the findings of this pilot RCT suggested that rESWT combined with physical therapy could be effective in relieving pain and improving function in muscle-related TMD patients (cite and refer to Marotta, N.; Ferrillo, M.; Demeco, A.; Drago Ferrante, V.; Inzitari, M.T.; Pellegrino, R.; Pino, I.; Russo, I.; de Sire, A.; Ammendolia, A. Effects of Radial Extracorporeal Shock Wave Therapy in Reducing Pain in Patients with Temporomandibular Disorders: A Pilot Randomized Controlled Trial. Appl. Sci. 2022, 12, 3821. https://doi.org/10.3390/app12083821)
References were well written.
